290 metagenome-assembled genomes from the Mediterranean Sea: a resource for marine microbiology

Tully Benjamin J. tully.bj@gmail.com 1
Sachdeva Rohan 2
Graham Elaina D. 2
Heidelberg John F. 1 2
1 Center for Dark Energy Biosphere Investigations, University of Southern California , Los Angeles , CA , USA
2 Department of Biological Sciences, University of Southern California , Los Angeles , CA , United States of America
Kyrpides Nikos
Electronic publication date: 2017 Jul 10
Publication date: 2017
Volume: 5
Electronic Location ID: e3558
Received 2017 May 9; Accepted 2017 Jun 19
Copyright: ©2017 Tully et al.
Copyright year: 2017
Copyright holder: Tully et al.
License: This is an open access article distributed under the terms of the Creative Commons Attribution License, which permits unrestricted use, distribution, reproduction and adaptation in any medium and for any purpose provided that it is properly attributed. For attribution, the original author(s), title, publication source (PeerJ) and either DOI or URL of the article must be cited.
License URL: https://creativecommons.org/licenses/by/4.0/

Keywords: Metagenomics, Metagenome-assembled genomes, Mediterranean sea, Bacteria, Archaea, Tara oceans

Funding: Center for Dark Energy Biosphere Investigations (C-DEBI) OCE-0939654 Funding was provided by the Center for Dark Energy Biosphere Investigations (C-DEBI) to BJT and JFH (OCE-0939654). The funders had no role in study design, data collection and analysis, decision to publish, or preparation of the manuscript.

==============================
The Tara Oceans Expedition has provided large, publicly-accessible microbial metagenomic datasets from a circumnavigation of the globe. Utilizing several size fractions from the samples originating in the Mediterranean Sea, we have used current assembly and binning techniques to reconstruct 290 putative draft metagenome-assembled bacterial and archaeal genomes, with an estimated completion of ≥50%, and an additional 2,786 bins, with estimated completion of 0–50%. We have submitted our results, including initial taxonomic and phylogenetic assignments, for the putative draft genomes to open-access repositories for the scientific community to use in ongoing research.

Introduction

Microorganisms are a major constituent of the biology within the world’s oceans and act as important linchpins in all major global biogeochemical cycles (Falkowski, Fenchel & DeLong, 2008). Marine microbiology is among the disciplines at the forefront in understanding how microorganisms respond to and impact local and large-scale environments. An estimated 1029 Bacteria and Archaea (Whitman, Coleman & Wiebe, 1998) reside in the oceans and represent an immense amount of poorly constrained, and ever evolving genetic diversity.

The Tara Oceans Expedition (2003–2010) was a major endeavor to add to the body of knowledge collected during previous global ocean surveys to sample the genetic potential of microorganisms (Karsenti et al., 2011). To accomplish this goal, Tara Oceans sampled planktonic organisms (viruses to fish larvae) at two major depths, the surface ocean and the mesopelagic. The amount of data collected was expansive and included 35,000 samples from 210 ecosystems (Karsenti et al., 2011). The Tara Oceans Expedition generated and publically released 7.2 Tbp of metagenomic data from 243 ocean samples from throughout the global ocean, specifically targeting the smallest members of the ocean biosphere, the viruses, Bacteria and Archaea, and picoeukaryotes (Sunagawa et al., 2015). Initial work on these fractions produced a large protein database, totaling >40 million nonredundant protein sequences and identified >35,000 microbial operational taxonomic units (OTUs) (Sunagawa et al., 2015).

Leveraging the publically available metagenomic sequences from the “girus” (giant virus; 0.22–1.6 µm), “bacteria” (0.22–1.6 µm), and “protist” (0.8–5 µm) size fractions, we have performed a new joint assembly of these samples using current sequence assemblers (Megahit (Li et al., 2016)) and methods (combining assemblies from multiple sites using Minimus2 (Treangen et al., 2011)). These metagenomic assemblies were binned using BinSanity (Graham, Heidelberg & Tully, 2017) into 290 draft microbial genomes with an estimated completeness ranging from 50–100%. Environmentally derived genomes are imperative for a number of downstream applications, including comparative genomics, metatranscriptomics, and metaproteomics. This series of genomic data can allow for the recruitment of environmental “-omic” data and provide linkages between functions and phylogenies. This method was initially performed on the seven sites from the Mediterranean Sea containing microbial metagenomic samples (TARA007, −009, −018, −023, −025 and −030), but will continue through the various Longhurst provinces (Longhurst et al., 1995) sampled during the Tara Oceans project (Fig. 1). All of the assembly data is publically available, including the initial Megahit assemblies for each site from the various size fractions and depths along with the recovered putative (minimal quality control) genomes.

Figure 1 Map illustrating the locations and size fractions sampled for the Tara Oceans Mediterranean Sea datasets.

Girus, ‘giant virus’ size fraction (0.22–1.6 µm). Bact, ‘bacteria’ size fraction (0.22–1.6 µm). Prot, ‘protist’ size fraction (0.8–5.0 µm). The map in Fig. 1 was modified under a CC BY-SA 3.0 license from ‘Blank Map of South Europe and North Africa’ by historicair (https://upload.wikimedia.org/wikipedia/commons/d/db/Blank_map_of_South_Europe_and_North_Africa.svg).

Figure 2 Workflow used to process Tara Oceans Mediterranean Sea metagenomic datasets.

Materials and Methods

A generalized version of the following workflow is presented in Fig. 2.

Sequence retrieval and assembly

All sequences for the reverse and forward reads from each sampled site and depth within the Mediterranean Sea were accessed from European Molecular Biology Laboratory-European Bioinformatics Institute (EMBL-EBI) utilizing their FTP service (Table 1). Paired-end reads from different filter sizes from each site and depth (e.g., TARA0007, girus filter fraction, sampled at the deep chlorophyll maximum) were assembled using Megahit (Li et al., 2016) (v1.0.3; parameters: -preset, meta-sensitive). To keep consistent with TARA sample nomenclature, “bacteria” or “BACT” will be used to encompass the size fraction 0.22–1.6 µm. Megahit assemblies ≥2 kb in length from all samples were pooled and combined based on ≥99% semi-global identity using CD-HIT-EST (Fu et al., 2012) (v4.6; -c 0.99) in order to reduce the number of redundant contigs for the downstream assembly step. The reduced set of contiguous DNA fragments (contigs) was then cross-assembled based on a minimum of 100 bp overlaps with 95% nucleotide identity using Minimus2 (Treangen et al., 2011) (AMOS v3.1.0; parameters: -D OVERLAP = 100 MINID = 95). This assembly method is available on Protocols.io at https://dx.doi.org/10.17504/protocols.io.hfqb3mw.

Table 1 Statistics for Megahit contigs, recruitment to data-rich-contigs, and relative abundance of draft genome results for each sample.

TARA sample site	Size fraction (Girus, Bacteria, or Protist)	Depth (Surface or DCMa)	No. of reads	No. of initial Megahit assembly	N50c (bp; initial Megahit assembly)	Longest initial Megahit assembly (bp)	Recruitment (% data-rich-contigs)	Relative abundancec of draft genomes (%)	Relative abundancec of ten most abundant genomes (% )	
TARA007	Girus	DCM	178,519,830	1,318,470	828	220,754	72.84	14.64	6.35	
TARA007	Girus	Surface	221,166,612	1,308,847	861	211,946	81.74	14.83	6.12	
TARA007	Protist	DCM	744,458,992	4,667,618	654	188,635	19.45	8.60	3.18	
TARA007	Protist	Surface	265,432,098	2,590,120	564	18,444	25.58	1.57	0.61	
TARA009	Girus	DCM	416,553,274	2,796,841	831	1,643,839	69.48	14.16	6.32	
TARA009	Girus	Surface	489,617,426	1,787,467	929	1,142,851	68.85	12.29	4.76	
TARA009	Protist	DCM	329,036,110	1,938,636	613	95,724	22.07	13.35	4.20	
TARA009	Protist	Surface	370,813,078	1,700,350	588	292,050	22.53	15.97	6.17	
TARA018	Bacteria	DCM	408,021,182	2,520,645	840	1,573,060	76.22	11.49	3.18	
TARA018	Bacteria	Surface	414,976,308	2,604,031	816	2,086,508	75.80	11.03	3.02	
TARA023	Bacteria	DCM	147,400,552	1,273,576	830	213,456	76.08	13.29	4.09	
TARA023	Bacteria	Surface	149,566,010	1,237,617	825	134,179	75.98	13.82	4.01	
TARA023	Protist	DCM	508,610,652	2,707,801	734	336,689	28.23	25.07	7.83	
TARA023	Protist	Surface	397,044,232	2,246,571	593	397,140	23.00	25.16	10.31	
TARA025	Bacteria	DCM	386,627,816	2,516,865	806	388,546	69.77	14.55	5.35	
TARA025	Bacteria	Surface	457,560,422	2,326,838	857	330,773	75.57	10.99	3.18	
TARA030	Bacteria	DCM	346,837,034	1,968,945	1,097	508,775	80.16	10.31	2.57	
TARA030	Bacteria	Surface	478,785,582	1,639,697	1,194	204,976	77.70	7.26	2.64	
TARA030	Protist	DCM	426,896,616	1,620,343	616	478,892	15.12	17.83	5.13	
TARA030	Protist	Surface	430,029,974	1,838,588	628	287,782	22.36	17.60	6.73	
Notes.

a DCM—deep chlorophyll maximum.

b N50—length of DNA sequence above which 50% of the total is contained.

c Relative abundance—determined using the reads recruited data-rich-contigs.

Metagenome-assembled genomes

Due to memory limitations during the binning step, contigs ≥7.5 kb in length generated during the two-step assembly process (includes Minimus2 contigs and unincorporated Megahit contigs) were used to recruit sequence reads from each of the Tara samples using Bowtie2 (Langmead & Salzberg, 2012) (v4.1.2; default parameters). Read counts for each contig were determined using featureCounts (Liao, Smyth & Shi, 2014) (v1.5.0; default parameters). Coverage was determined for all contigs by dividing the number of recruited reads by the length of the contig (reads/bp). Due to the low coverage nature of the samples, in order to effectively delineate between contig coverage patterns, the coverage values were transformed by multiplying by five (determined through manual tuning). Transformed coverage values were then utilized to cluster contigs into bins utilizing BinSanity (Graham, Heidelberg & Tully, 2017) based on a preference value of −3 to be run for a maximum of 4,000 iterations with completion if convergence is achieved for 400 consecutive iterations and a damping value of 0.9 (parameters: -p -3, -m 4,000, -v 400, -d 0.9). Bins were assessed for the presence of putative microbial genomes using the lineage workflow in CheckM (Parks et al., 2015) (v1.0.3; parameters: lineage_wf). Bins were split in to three categories: (1) putative draft genomes (≥50% complete and ≤10% cumulative redundancy (% contamination—(% contamination × % strain heterogeneity ÷ 100))); (2) draft genomes with high contamination (≥50% complete and ≥10% cumulative redundancy); and (3) low completion bins (<50% complete).

The high contamination bins containing approximately two genomes, three genomes, or ≥4 genomes used BinSanity (Binsanity; -m 2000, -v 200, -d 0.9) with variable preference values (-p) of −1,000, −500, and −100, respectively. The resulting bins were added to one of the three categories: putative draft genomes, high contamination bins, and low completion bins. The high contamination bins were processed for a third time with the Binsanity-refine utilizing a preference of −100 (-p −100). These bins were given final assignments as either putative draft genomes or low completion bins.

Any contigs not assigned to putative draft genomes were assessed using BinSanity using raw coverage values. Two additional rounds of refinement were performed with the first round of refinement using preference values based on the estimated number of contaminating genomes (as above) and the second round using a set preference of −10 (-p −10). Following this binning phase, contigs were assigned to draft genome bins (e.g., Tara Mediterranean genome 1, referred to as TMED1, etc.), low completion bins with at least five contigs (0–50% complete; TMEDlc1, etc. lc, low completion), or were not placed in a bin (Supplemental Informations S1 and S2).

Taxonomic and phylogenetic assignment of draft genomes

The bins representing the draft genomes were assessed for taxonomy and phylogeny using multiple methods to provide a quick reference for selecting genomes of interest. Taxonomy was assigned using the putative placement provided via CheckM during the pplacer (Matsen, Kodner & Armbrust, 2010) step of the analysis to the lowest taxonomic placement (parameters: tree_qa -o 2). This step was also performed for all low completion bins.

Figure 3 FastTree approximate maximum-likelihood phylogenetic tree constructed with 37 and 406 16S rRNA genes from putative draft genomes and references, respectively.

Sequence alignment is available in Supplemental Information 4. Phylogenetic tree with Shimodaira–Hasegawa local support values as available in Newick format in Supplemental Information 5.

Two separate attempts were made to assign the draft genomes a phylogenetic assignment. Draft genomes were searched for the presence of the full-length 16S rRNA gene sequence using RNAmmer (Lagesen et al., 2007) (v1.2; parameters: -S bac -m ssu). All full-length sequences were aligned to the SILVA SSU reference database (Ref123) using the SINA web portal aligner (Pruesse, Peplies & Glöckner, 2012) (https://www.arb-silva.de/aligner/). These alignments were loaded in to ARB (Ludwig et al., 2004) (v6.0.3), manually assessed, and added to the non-redundant 16S rRNA gene database (SSURef123 NR99) using ARB Parsimony (Quick) tool (parameters: default). A selection of the nearest neighbors to the Tara genome sequences were selected and used to construct a 16S rRNA phylogenetic tree. Genome-identified 16S rRNA sequences and SILVA reference sequences were aligned using the SINA web portal aligner (Supplemental Information 4). An approximately-maximum-likelihood tree with Shimodaira–Hasegawa local support values (Shimodaira & Hasegawa, 1999) was constructed using FastTree (Price, Dehal & Arkin, 2010) using the generalized time reversible and discrete gamma models (Yang) (v2.1.3; parameters: -nt -gtr -gamma; Fig. 3; Supplemental Information 5).

Draft genomes were assessed for the presence of the 16 ribosomal markers genes used in Hug et al. (2016). Putative CDSs were determined using Prodigal (v2.6.3; parameters: -m -p meta) and were searched using HMMs for each marker using HMMER (Finn, Clements & Eddy, 2011) with matches based on an e-value cutoff of 1 × 10−3 (v3.1b2; parameters: hmmsearch -E 1E −10). If a genome had multiple copies of any single marker gene, neither was considered, and only genomes with ≥8 markers were used to construct a phylogenetic tree. Markers identified from the draft genomes were combined with markers from 6,080 reference genomes accessed from NCBI GenBank (Supplemental Information 6) that represent the major bacterial phylogenetic groups. Each marker gene was aligned using MUSCLE (Edgar, 2004) (parameters: -maxiters 8) and automatically trimmed using trimAL (Capella-Gutiérrez, Silla-Martínez & Gabaldón, 2009) (v1.2rev59; parameters: -automated1). Automated trimming results were manually curated in Geneious (Kearse et al., 2012) (Supplemental Information 7). Final alignments were concatenated and used to construct an approximately-maximum-likelihood tree using the LG (Le & Gascuel, 2008) and Gamma models with Shimodaira–Hasegawa local support values with FastTree (v2.1.10; parameters: -lg -gamma; Fig. 4; Supplemental Information 8). A separate tree was constructed using the same concatenated alignments and tree-building parameters for the 210 draft genomes, without the reference genomes (Fig. 4).

Figure 4 Cladogram of a FastTree approximate maximum-likelihood phylogenetic tree constructed using 16 syntenic, single-copy marker genes for 210 draft genomes.

Support for internal nodes were determined based on the Shimodaira–Hasegawa test (white ≥0.500, gray ≥0.750, black ≥0.950). Sequence alignment for the draft genomes and the reference genomes used for phylogenetic assignment is available in Supplemental Information 7. Phylogenetic tree of the draft genomes and the reference genomes used for phylogenetic assignment is available in Newick format in Supplemental Information 8.

Relative abundance of draft genomes

To set-up a baseline that could approximate the “microbial” community (Bacteria, Archaea and viruses) present in the various Tara metagenomes, which included filter sizes specifically targeting both protists and giruses, reads were recruited against all contigs generated from the Minimus2 and Megahit assemblies ≥2 kb using Bowtie2 (default parameters). The contigs <2 kb in length likely constitute low abundance bacteria and archaea, bacteria and archaea with high degrees of repeats resulting in poor assembly, fragmented picoeukaryotic genomes, and problematic read sequences (low quality, sequencing artefacts, etc.) and were not included in further analysis. All relative abundance measures are relative to the number of reads recruited to the assemblies ≥2 kb. Read counts were determined using featureCounts (as above). Length-normalized relative abundance values were determined for each draft genome for each sample: Readsbppergenome∑Readsbpallgenomes×∑Recruitedreadstogenomes∑Recruitedreadstoallcontigs≥2kb×100.

Results

Assembly

The initial Megahit assembly was performed on the publicly available reads for Tara stations 007, 009, 018, 023, 025, 030. Starting with 147–744 million reads per sample, the Megahit assembly process generated 1.2–4.6 million contigs with a mean N50 and longest contig of 785 bp and 537 kb, respectively (Table 1). In general, the contigs generated from the Tara samples targeting the protist size fraction (0.8–5 µm) had a shorter N50 value than the bacteria size fractions (mean: 554 bp vs 892 bp, respectively). Contigs from the Megahit assembly process were pooled and separated by length. Of the 42.6 million contigs generated during the first assembly, 1.5 million were ≥2 kb in length (Table 2). Several attempts were made to assemble the shorter contigs, but publicly available overlap-consensus assemblers (Newbler (454 Life Sciences), cap3 (Huang & Madan, 1999), and MIRA (Chevreux et al., 2004)) failed on multiple attempts. Processing the ≥2 kb contigs from all of the samples through CD-HIT-EST reduced the total to 1.1 million contigs. This group of contigs was subjected to the secondary assembly through Minimus2, generating 158,414 new contigs (all ≥2 kb). The secondary contigs were combined with the Megahit contigs that were not assembled by Minimus2. This provided a contig dataset consisting of 660,937 contigs, all ≥2 kb in length (Table 2; further referred to as data-rich-contigs).

Binning

The set of data-rich-contigs was used to recruit the metagenomic reads from each sample using Bowtie2. The data-rich-contigs recruited 15–81% of the reads depending on the sample. In general, the protist size fraction recruited substantially fewer reads than the girus and bacteria size fractions (mean: 19.8% vs 75.0%, respectively) (Table 1). For the protist size fraction, the “missing” data for these recruitments likely results from the poor assembly of more complex and larger eukaryotic genomes. The fraction of the reads that do not recruit in the girus and bacterial size fraction samples could be accounted for by the large number of low quality assemblies (200–500 bp) and reads that could not be assembled due to low abundance or high complexity (Table 2).

Table 2 Assembly statistics at various steps during processing.

Contig grouping	No. of contigs	N50a	Total sequence (bp)	
Megahit assemblies 200–499 bp	24,999,285	n.d.	9,293,098,676	
Megahit assemblies 500–1,999 bp	16,103,221	n.d.	13,382,057,993	
Megahit assemblies ≥2 kb	1,517,360	4,658	6,691,877,664	
Megahit assemblies ≥2 kb (post-CD-HIT-EST)	1,126,975	4,520	4,894,479,496	
Minimus2 contigs	158,414	15,394	1,727,079,865	
Minimus2 + unincorporated Megahit contigs ≥2 kb (data-rich-contigs)	660,937	5,466	3,612,405,904	
Minimus2 + unincorporated Megahit contigs ≥7.5 kb (binned-contigs)	95,506	20,556	1,725,063,313	
Notes.

a N50—length of DNA sequence above which 50% of the total is contained.

Unsupervised binning was performed using both transformed and raw coverage values for a subset of 95,506 contigs from the data-rich-contigs that were ≥7.5 kb (referred to further as binned-contigs). Binning using the transformed coverage data generated 237 putative draft genomes containing 15,032 contigs (Supplemental Information 1). Contigs not in putative genomes were re-binned based on raw coverage values, generating 53 additional putative draft genomes encompassing 3,348 contigs. In total, 290 putative draft genomes were generated with 50–100% completion (mean: 69%) with a mean length and number of putative CDS of 1.7Mbp and 1,699, respectively (Supplemental Information 1). In analyzing the quality of generated draft genomes, 31 of the genomes had a contamination value >10%, but had a cumulative redundancy value <10%, while an additional 12 genomes had a cumulative redundancy value >10% (genomes highlighted in Supplemental Information 1). For instances where the predicted strain heterogeneity is high and sharply reduces the calculated cumulative redundancy value (e.g., TMED1), downstream analysis of these genomes may offer opportunities to examine within strain variation, if the gene content varies between contigs, or identify duplicate contigs that can be removed to rectify contamination issues. For instances where cumulative redundancy values remain high (e.g., TMED20 or TMED106), downstream analysis utilizing composition signatures, for example within Anvi’o (Eren et al., 2015), should be able to identify problematic contigs and reduce the overall contamination reported for the genome.

All other contigs were grouped into bins with at least five contigs, but with estimated completion of 0–50% (2,786 low completion bins; 74,358 contigs; Supplemental Information 2) or did not bin (2,732 contigs). Nearly a quarter of the 2,786 low completion bins (24.7%) have an estimated completion of 0%. These bins may be good candidates for exploring small double-stranded viral genomes or episomal genetic elements.

Taxonomy, phylogeny, & potential organisms of interest

The 290 putative draft genomes had a taxonomy assigned to it via CheckM during the pplacer step. All of the genomes, except for 20, had an assignment to at least the Phylum level, and 83% of the genomes had an assignment to at least the Class level (Supplemental Information 1).

Phylogenetic information was determined for as many genomes as possible. Genomes were assessed for the presence of full-length 16S rRNA genes. In total, 37 16S rRNA genes were detected in 35 genomes. 16S rRNA genes can prove to be problematic during the assembly steps due to the high level of conservation that can break contigs (Miller, Koren & Sutton, 2010) (Fig. 3). The conserved regions of the 16S rRNA, depending on the situation, can over- or under-recruit reads, resulting in coverage variations that can misplace contigs into the incorrect genome. As such, most of the 16S rRNA phylogenetic placements support the taxonomic assignments, while eight of the assignments were contradictory in nature (denoted in Supplemental Information 1). An example of this nonconformity of 16S rRNA assignment would be TMED32. A Bacteroidetes, TMED32 is assigned to the Order Cytophagales via CheckM and the ribosomal marker tree and contains three 16S rRNA sequences. One of the 16S rRNA sequences is conformational, with placement in the Family Flammeovirgaceae, while the remaining two 16S rRNA are assigned to the Mitochondria. For future research purposes, contigs with contradictory 16S rRNA or incongruent phylogenetic/taxonomic signatures should be removed. Downstream analysis should allow for the determination of the most parsimonious result in any draft genome with contradictory phylogenetic assignments.

Beyond the 16S rRNA gene, genomes were searched for 16 conserved, syntenic ribosomal markers. Sufficient markers (≥8) were identified in 210 of the genomes (72%) and placed on a tree with 6,080 reference sequences and used to assign a putative phylogeny (Supplemental Information 1). Phylogenies were then assigned to the lowest taxonomic level that could be confidently determined (Fig. 4). These putative results reveal a number of genomes were generated that represent multiple clades for which environmental genomic information remains limited, including: Planctomycetes, Verrucomicrobia, Cyanobacteria, and uncultured groups within the Alpha- and Gammaproteobacteria.

Relative abundance

A length-normalized relative abundance value was determined for each genome in each sample based on the number of reads recruited to the data-rich-contigs. The relative abundance for the individual genomes was determined based on this portion of the dataset (Supplemental Information 3). In general, the genomes had low relative abundance (maximum relative abundance = 1.9% for TMED155 a putative Cyanobacteria at site TARA023 from the protistan size fraction sampled at the surface; Supplemental Information 1). The draft genomes accounted for 1.57–25.16% of the approximate microbial community as determined by the data-rich-contigs (mean = 13.69%), with the ten most abundant genomes in a sample representing 0.61–10.31% (Table 1).

Concluding Statement

The goal of this project was to provided preliminary putative genomes from the Tara Oceans microbial metagenomic datasets. The 290 putative draft genomes and 2,786 low completion bins were created using the 20 samples and six stations from the Mediterranean Sea.

Initial assessment of the phylogeny of these metagenomic-assembled genomes based on concatenated ribosomal markers indicates several new genomes from environmentally relevant organisms, including approximately 10 new Cyanobacteria genomes within the genera Prochlorococcus and Synechococcus and 22 new SAR11 genomes. Additionally, there are putative genomes from the marine Euryarchaeota (n = 13), Verrucomicrobia (n = 15), and Planctomycetes (n = 7). Additionally, the low completion bins may house distinct viral genomes. Of particular interest may be the 40 bins with 0% completion (based on single-copy marker genes), but that contain >500 kb of genetic material (including 3 bins with >1 Mb). These large bins lacking markers may be good candidates for research in to the marine “giant viruses” and episomal DNA sources (plasmids, etc.).

It should be noted, researchers using this dataset should be aware that all of the genomes generated from these samples should be used as a resource with some skepticism towards the results being an absolute. Like all results for metagenome-assembled genomes, these genomes represent a best-guess approximation of a taxon from the environment (Sharon & Banfield, 2013). Researchers are encouraged to confirm all claims through various genomic analyses and accuracy may require the removal of conflicting sequences.

Supplemental Information

Supplemental Information 1 Statistics, taxonomic and phylogenetic assignments for the putative high-quality genomes

Click here for additional data file.

Supplemental Information 2 Statistics and CheckM taxonomy for low completion bins

Click here for additional data file.

Supplemental Information 3 Relative abundance values determined for each genome based the length-normalized fraction of reads recruited to the genome relative to reads recruited for the data-rich-contigs

Click here for additional data file.

Supplemental Information 4 SINA alignment of 16S rRNA sequences from the 37 draft genomes and 406 reference sequences

Click here for additional data file.

Supplemental Information 5 Newick file of the 16S rRNA tree used to generated Fig. 3. Contains Shimodaira-Hasegawa local support values

Click here for additional data file.

Supplemental Information 6 Information about the 6,080 reference genomes used to generate the 16 ribosomal marker protein tree

Click here for additional data file.

Supplemental Information 7 Concatenated MUSCLE alignment file of 16 ribosomal marker proteins used to determine phylogeny for the draft genomes utilizing 6,080 reference genomes

Click here for additional data file.

Supplemental Information 8 Newick file of concatenated 16 ribosomal marker proteins, including FastTree determined local support values using the Shimodaira-Hasegawa test for draft and reference genomes

Click here for additional data file.

We are indebted to the Tara Oceans project and team for their commitment to open-access data that allows data aficionados to indulge in the data and attempt to add to the body of science contained within. This is C-DEBI Contribution 333.

Additional Information and Declarations

Competing Interests

Author Contributions

Data Availability

The authors declare there are no competing interests.

Benjamin J. Tully conceived and designed the experiments, performed the experiments, analyzed the data, wrote the paper, prepared figures and/or tables, reviewed drafts of the paper.

Rohan Sachdeva reviewed drafts of the paper, provided origins of assembly workflow.

Elaina D. Graham performed the experiments, reviewed drafts of the paper.

John F. Heidelberg contributed reagents/materials/analysis tools, reviewed drafts of the paper.

The following information was supplied regarding data availability:

This project has been deposited at DDBJ/ENA/GenBank under the BioProject accession no. PRJNA385857 and drafts of genomes are available with accession no. NHBG00000000– NHMJ00000000. All contigs generated using Megahit from each sample are available through iMicrobe (http://data.imicrobe.us/project/view/261). TARA Oceans has deposited the reads in the NCBI Sequence Read Archive (SRA) with accessions ERS488346, ERS488330, ERS477998, ERS477979, ERS488509, ERS488486, ERS478040, ERS477953, ERS477931, ERS488147, ERS488119, and ERS478017.

Additional files have been provided and are available through FigShare, such as: all contigs from Minimus2 + Megahit output used for binning and community assessment; contig read counts per sample; the putative genome contigs and Prodigal-predicted nucleotide and protein putative CDS FASTA files; the ribosomal marker HMM profiles; reference genome markers; draft genome markers; low completion bins, and contigs without a bin.

Tully, Benjamin; Sachdeva, Rohan; Graham, Elaina; Heidelberg, John (2017): 290 Genomes from the Mediterranean Sea: Supplemental Data. figshare. https://doi.org/10.6084/m9.figshare.3545330.v3.

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
