# Peer review of "metagenome-assembled genomes from the Mediterranean Sea: a resource for marine microbiology"

_PeerJ, doi:10.7717/peerj.3558_

## Round 0.1 · original submission · Minor Revisions

The manuscript has been reviewed by three expert reviewers. They all agree that the data presented will provide a very useful resource to the community and most of the requested changes seem to be minor. However two of the reviewers also raised concerns for the definitions of the quality of the assembled genomes, suggesting higher quality scores, which should be addressed in the revised version

·

Basic reporting

The authors present a workflow from recovering metagenome-assembled genomes that utilizes Megahit, Minimus2, BinSanity, and CheckM. This pipeline is applied to metagenomic samples from around the Mediterranean Sea and the 290 genomes recovered from these metagenomes presented as a genomic resource for use by the larger research community. I agree that these genomes will be a useful resource to research groups interested in marine systems and microbial diversity more generally. I do have some concerns and questions regarding the workflow, the reported quality of the metagenome-assembled genomes, and the presented phylogenetic assignments.

1. Overall, the manuscript reads well, but it would benefit for additional proofreading. A few examples include:
Line 25: “as the important linchpins” would be easier to read as “as important linchpins”
Line 26: “at the forefront of advancements in understanding” is awkward to read
Line 27: “impact the local and large-scale environments” would be easier to read as “impact local and large-scale environments”
Line 30: “encompassed a major endeavor” would be easier to read as “was a major endeavor”
Line 45: “genomes, ranging from 50-100% estimated completion” would be easier to read as “genomes with an estimated completeness ranging from 50-100%.”
Line 53: “depths and putative (minimal quality control) genomes” would be easier to read as “depths along with the recovered putative (minimal quality control) genomes”
Line 109: “Taxonomy as assigned…” should be “Taxonomy was assigned…”
Lines 147-149: I find the sentence on these lines hard to parse.

2. I found the phrasing “unassembled Megahit contigs” in Figure 2 to be a bit confusing. Perhaps something like “unmerged Megahit contigs” would be clearer?

Experimental design

1. It would be helpful to explicitly explain the parameter settings used for each program used in the presented workflow. For example, what do the “-c” and “-T” parameters of CD-HIT-EST control and why were these settings used. I appreciate that the parameter settings were likely set to achieve a desired general outcome (e.g, set to only cluster highly similar sequences) and I do not expect a detailed justification for the exact value used. It would also help clarify the workflow to explicitly indicate that the CD-HIT-EST step is done, I presume, purely for computational reasons.

2. Minimus2 was used to merge contigs from independent assemblies. I find this to be an interesting idea, but am concerned it may create some chimeric contigs. If I understand correctly, the parameters used (-D OVERLAP=100; MINID=95) will merge together any contigs that overlap by just 100 bp with 95% identity. Is there any concern that this is creating chimeric contigs? Was any formal or informal testing performed to ensure chimeric contigs are not being created (e.g., looking at the coverage across merged contigs)?

3. Please verify that the concatenated ribosomal protein tree was performed using genes in nucleotide space as suggested by running FastTree under the GTR model. If protein sequences were used, please indicate if FastTree was run with the JTT, WAG, or LG model.

Validity of the findings

1. High-quality genomes are defined in the manuscript as being ≥50% complete with ≤10% redundancy. A higher standard is generally being used when attributing the label “high quality”. For example, the CheckM manuscript describes high quality as being ≥90% complete and low contamination as ≤5% (Parks et al., 2015). Haroon et al. (Sci Data, 2016) highlight genomes where the (completeness – contamination) ≥50 and do not refer to these as high quality. The Human Microbiome project also uses a ≥90% completeness criterion for defining high-quality draft genomes (http://hmpdacc.org/reference_genomes/reference_genomes.php) and there is a forthcoming perspective paper suggesting that ≥90% completeness and ≤5% contamination be used to define high-quality metagenome-assembled genomes. I suggest referring to the 290 genomes that are ≥50% complete with ≤10% redundancy simple as “draft genomes”.

2. Redundacy is defined as [% contamination – (% contamination × % strain heterogeneity ÷ 100)]. What is the rational for using this measure compared to the estimated contamination for filtering genomes? I believe the use of this measure makes the genomes appear “cleaner” than they are as strain heterogeneity can be problematic given the variability often observed between strains. Specifically, the 20 genomes with ≥15% contamination are of concern, especially as some of these have <<100% strain heterogeneity (e.g., TMED20 is reported as having ~41% contamination and ~76% strain heterogeneity, suggesting it still has 10% “non-strain” contamination). My personal view would be to filter out all genomes that have ≥10% contamination regardless of the estimated level of strain heterogeneity or, more rigorously, to apply the same criteria as Haroon et al. (Sci Data, 2016). If genomes with high levels of contamination due to a mixture of strains are to be retained this needs to be more clearly stated in the manuscript.

3. There are 12 genomes in Supp. Table 1 explicitly indicated as having ≥10% redundancy. Why are these included as part of the 290 “high quality” genomes? The presence of these is clearly stated, but it is unclear why there are retained as “high quality” genomes given they don’t meet the specified inclusion criteria.

4. Several of the genomes with 16S rRNA genes are classified as “g__Mitochondria” or “g__Chloroplast”. It is unclear to me what these refer to in terms of bacterial taxonomy.

5. Given the authors are presenting the provided genomes as a genomic resource with taxonomic and phylogenetic assignments, it would seem appropriate to try and resolve the identified incongruent placements between the 16S rRNA and concatenated ribosomal protein trees. If the 16S can be determined to be erroneously binned (as suggested), should they not be removed?

6. What is the agreement between the CheckM classifications and those provided by the concatenated ribosomal protein tree?

Additional comments

1. Line 170: Should “1.2-4.6 million assemblies” be “1.2-4.6 million assembled contigs”?

2. The Supplemental Material is sometimes referred to as “Supplemental Information” and other times as “Supplemental Table”.

Reviewer 2 ·

Basic reporting

The manuscript by Tully et al., is a well-conducted study that provides a genomic dataset for other scientist by assembling and binning new genomes from the Tara Oceans project. Overall, the authors managed to report the objective of their study, the employed methodology and results well throughout the manuscript. Additionally, all data was made available and was presented in a clear and readable manner.

A few comments on parts of the manuscript that could have been explained a bit more extensively:

1. Line 49: What was the objective when choosing these seven sites. While I do understand that it is not the objective of this study to work with the whole dataset, it would be interesting to include why these specific locations were chosen.

2. Line 135: How many bacterial and archaeal genomes were reconstructed. Were any sequences assigned to eukaryotes assembled?

3. Line 219: What is the exact number of cases, where the 16S phylogeny contradicted the ribosomal protein phylogeny or the placement using CheckM?

Experimental design

The authors present the experimental design in a clear and logical way and the usefulness of this dataset was explained sufficiently.

There are just some minor points, where the methods could have been stated more clearly:

1. Line 40-41: There is no size difference for the girus or bacteria fraction; therefore, it is unclear to me how these fractions were named?

2. Line 63: Was some initial quality control done on the raw reads to remove low quality reads?

3. In the section “sequence retrieval and assembly”: It is not completely clear to me if at one point the individual assemblies were combined in a co-assembly. For example when stating ‘all of the assemblies were pooled’; does this mean all samples were combined for a co-assembly in the second part of the assembly pipeline?

4. Line 75: Read mapping was done against contigs >7.5 kb. Does this mean that the binning was also performed with contigs >7.5 kb?

Validity of the findings

The data appears robust and at whenever necessary, the authors did point out the limitations of their work in a convincing way (i.e. the problematic of miss-assigning 16S sequences).

Additional comments

1. Line 62: I was not able to find the raw data quickly; to make the raw reads more accessible the authors could consider adding the website link or the EMBL ID.

2. Line 96: There seems to be something missing at ‘…, but have been designated.’

3. Supplemental Information S1: It is not clear to me what “cumlative contamination” means. For example TMED1 has a contamination of 13.64%, however, it is not marked with a star. Also there is a typo in ‘cumlative’.

Reviewer 3 ·

Basic reporting

Tully et al. provide a dataset of metagenome-assembled genomes (MAGs) from the Mediterranean subset of the larger TARA Oceans Expedition. The authors undertook an iterative assembly method and binning effort, and provide limited analysis in the form of taxonomic assignment for 290 putative “high-quality” MAGs. While the data will undoubtedly be useful for the larger research community, I would highly recommend the authors carefully evaluate the data for quality.

Experimental design

The authors should be commended in their efforts for reproducibility and scientific openness. It was refreshing to see the full assembly pipeline was made available through protocols.io and and the methods were well described. I would suggest the authors go even further to enable full reproducibility of their work by including additional supplementary tables that list the reference genome sequences from IMG used in the analysis. What were the 1,729 reference genomes from IMG? A table listing those genomes should be included in the supplementary information.

A soon-to-be published standards developed by the Genomic Standards Consortium (GSC) has outlined minimum information about metagenome-assembled genomes (MIMAG) and single amplified genomes (MISAG). While the authors are not privy to these community-accepted standards yet (publication in press), I would highly recommend they attempt to conform to the standards as follows for high-quality draft: > 90% estimated genome completion along with the presence of the 23S, 16S and 5S, at least 18 tRNAs and < 5% estimated contamination. Otherwise, I suggest the authors modify the text to indicate that the 290 genomes are not high-quality but instead medium quality (50-90% complete, <10% contamination). Irrespective of the official GSC standards, I would certainly not consider 50% “high quality” as the authors indicate.

Lines 156-157: The authors list the accessions as: “This project has been deposited at DDBJ/ENA/GenBank under the BioProject accession no. #### and drafts of genomes are available with accession no. #####-#####.” This reviewer cannot adequately review the submitted manuscript without access to the submitted data at NCBI. Further, the link to the data via iMicrobe appears to be broken: http://imicrobe.us/project/view/261

Validity of the findings

I can appreciate the concluding statement and caveats for the MAGs presented. Given this statement, the authors should not attempt to submit the low-quality bins as “references” to NCBI. These data can be hosted on FigShare or other resources, but depositing low-quality MAGs to NCBI would further pollute the continually growing database of poor quality genomes.

Additional comments

Below are further suggestions for the authors to consider to make the data more widely useful to the community.

Line 205: The statement is a bit nonsensical: “Nearly a quarter of the low completion bins (24.7%) have an estimated completion of 0%.” Why would these be considered bins? I recommend the authors focus their analysis only on the high quality bins and not attempt to provide further information on “bins” that are estimated to have no estimated completeness at all. The distinction between 0% complete bins and the ~2700 contigs without bins should be made explicit if the authors are going to indicate that bins have 0% complete.

The discussion in lines 213-221 regarding challenges in assembly and binning of 16S is accurate. The authors should also indicate that in addition, 16S sequences typically have different %GC compared to the rest of the genome and therefore do not bin well. This is perhaps more important than the coverage differences the authors note. Further, the authors found 37 16S in 35 genomes and surprisingly there was one genome (TMED32) with 3 “16s” – two mitochondrial sequences and what looks to be an authentic Bacteroidetes 16S. Would recommend the authors highlight this in the main text.

The analysis for relative abundance based on length normalization is confusing. Why didn’t the authors simply use coverage to determine relative abundances for all datasets?

TARA data was retrieved from EMBL-EBI, not just EMBL.

Figure 3 should include bootstrap values. Also, the resolution is not great, but hopefully the authors use the NCBI accession number to label the references sequences.

Figure 4 lists Deltaproteobacteria, Chloroflexi and Firmicutes in parentheses?


It is surprising that “high quality” bins were not obtained for the archaea. Could the authors speculate as to why the Archaea was not better represented in their binning exercise?

Supplementary Tables 1 and 2 should include additional basic statistics for the binned genomes, including N50, L50, largest contig, number of contigs, percent of reads that map to assembly. Without this information, the quality of these genome bins cannot be accurately assessed.

The 16S alignments using MUSCLE is questionable as this is not the most appropriate aligner for 16S sequences. Why were the SINA alignments not used for tree reconstruction? These are more accurate alignments compared to MUSCLE. Alternatively cmalign is a superior method that takes into account the structure of the 16S in addition to sequence similarity. The authors should construct their phylogenies using an alignment algorithm other than MUSCLE that is robust to 16S.

Line 46: Text should be updated from “including comparative genomes” to “including comparative genomics.”

Line 79-85: The transformation method applied to the contig coverage appears arbitrary and not properly validated. Why would multiplying coverage by 5 “..determined by manual tuning…” improve the binning compared to the true coverage values? Is this a method widely used by other binning algorithms or is this idiosyncratic to BinSanity? I suggest the authors spend some effort validating this seemingly arbitrary data transformation. Perhaps comparing to other binning methods that are more widely used (eg. GroopM, MaxBin, Concoct, Metabat).

---

## Round 0.2 · accepted · Accept

Thank you for addressing all the reviewers comments. The manuscript has now been significantly improved and should be accepted as is.